# Validity of the Central Sensitization Inventory to Address Human Assumed Central Sensitization: Newly Proposed Clinically Relevant Values and Associations

**DOI:** 10.3390/jcm12144849

**Published:** 2023-07-23

**Authors:** Ingrid Schuttert, André P. Wolff, Rita H. R. Schiphorst Preuper, Alec G. G. A. Malmberg, Michiel F. Reneman, Hans Timmerman

**Affiliations:** 1Pain Center, Department of Anaesthesiology, University Medical Center Groningen (UMCG), University of Groningen, 9713 GZ Groningen, The Netherlands; i.schuttert@umcg.nl; 2Department of Rehabilitation Medicine, University Medical Center Groningen (UMCG), University of Groningen, 9713 GZ Groningen, The Netherlands; h.r.schiphorst.preuper@umcg.nl (R.H.R.S.P.); m.f.reneman@umcg.nl (M.F.R.); 3Department of Obstetrics and Gynaecology, University Medical Center Groningen (UMCG), University of Groningen, 9713 GZ Groningen, The Netherlands; g.g.a.malmberg@umcg.nl

**Keywords:** chronic pain, central sensitization inventory (CSI), central sensitization (CS), sensitization, central sensitivity syndrome (CSS), human assumed central sensitization (HACS), validity, cutoff value, sensitivity, specificity

## Abstract

Central sensitization cannot be directly demonstrated in humans and thus a gold standard is missing. Therefore, we used human assumed central sensitization (HACS) when associated with humans. The central sensitization inventory (CSI) is a screening questionnaire for addressing symptoms that are associated with HACS. This cross-sectional study compared patients with chronic pain and at least one central sensitivity syndrome with healthy, pain-free controls via ROC analyses. Analyses were performed for all participants together and for each sex separately. Regression analyses were performed on patients with chronic pain with and without central sensitivity syndromes. Based on 1730 patients and 250 healthy controls, cutoff values for the CSI for the total group were established at 30 points: women: 33 points; men: 25 points. Univariate and multivariate regression analyses were used to identify possible predictors for the CSI score in 2890 patients with chronic pain. The CSI score is associated with all independent factors and has a low association with pain severity in women and a low association with pain severity, age, and body mass index in men. The newly established CSI cutoff values are lower than in previous studies and different per sex, which might be of clinical relevance in daily practice and importance in research.

## 1. Introduction

Central sensitization (CS) is defined as the “Increased responsiveness of nociceptive neurons in the central nervous system to their normal or subthreshold afferent input [1]”. CS is suggested to increase the likelihood of pain becoming chronic [2]. The use of the term CS in clinical practice is under debate because it is only demonstrated in animal studies. It is difficult to distinguish between patients with and without CS based on objective pathophysiological mechanisms [3,4]. A gold standard to assess the presence and severity of CS is not available. Therefore, it has been previously suggested that CS should be described as human assumed central sensitization (HACS) if signs associated with CS are present in humans [5].

The central sensitization inventory (CSI) [6] was developed as a screening instrument to address experienced symptoms that are supposed to be associated with HACS [6]. The CSI is translated and validated in more than forty languages [7], among them the CSI-Dutch language version (SI) [8,9]. To address symptoms of HACS in patients with chronic pain, a cutoff point of 40 out of 100 was established [10]. The previously suggested cutoff scores for patients with various chronic pain disorders range between 11/100 [11] and 40/100 [10,12,13,14]. It is hinted that there might be a need for alternative cutoff values that may differ between different language versions/countries and between patient populations [15]. Until now, there is no specific cutoff value established for the CSI-Dutch language version (Dlv).

It is beneficial to address the presence of HACS in patients with chronic pain because it might affect the choice of an appropriate treatment to reduce HACS and to reduce the chronification of pain [16,17].

Women experience more pain and have a higher pain severity compared to men [18,19,20,21]. Sex differences might extend to the presentation and severity of HACS, and thus influence the estimation of the attribution to HACS and treatment effects. The number of central sensitization syndromes (CSSs) has been reported to be different between men and women with chronic musculoskeletal pain disorders [22]. The sex difference was also present when HACS was addressed with a CSI cutoff point of 40/100, but not when using the CSI total score [22]. Correlations between CSI and other factors also varied between studies. Pain catastrophizing correlated good [23,24], moderate [25,26,27] or low [15,28] with the CSI. Several disability questionnaires (SPADI [24], ODI [29], RMDQ [29,30], and NDI [31] correlated good [24,31] or moderately [29,30,31] with the CSI. The CSI correlated good [27,31], moderately [24,29,32,33] or low [23,34] with pain intensity, while age was weakly correlated with the CSI [15,33].

The primary aim of this study was to establish the cutoff values for the CSI-Dlv in patients with chronic pain. The secondary aims were (1) to address differences between groups (patients with chronic pain and healthy volunteers, males and females and the presence or absence of CSS) and (2) to address what factors are associated with the CSI score. 

## 2. Materials and Methods

The Medical Research Ethics Committee of the University Medical Center Groningen, Groningen, The Netherlands, granted an exemption for the patient study because it was based on patients’ standard medical files (METc 2020/284). An exemption was also provided for the inclusion of healthy volunteers because participation was not burdensome (METc 2021/361). The study was registered in the Dutch Trial Register under number NL9241 before the execution of the study. 

### 2.1. Participants

In this cross-sectional study, data were retrospectively collected from consecutive patients with chronic pain referred to the specialized multidisciplinary academic pain center at the University Medical Center Groningen (UMCG), Groningen, The Netherlands, between November 2017 and October 2021. 

The patients filled out the questionnaires online and sent them through RoQua (UMCG, Groningen, The Netherlands), a questionnaire program built into the electronic patient file (EPIC systems, Verona, WI, USA) before the first visit. Patients were included when they completed the CSI part A and part B and were aged ≥18 years. Patients of which sex and gender were not equal were excluded. A control group of healthy volunteers aged ≥18 years was recruited between July 2021 and October 2021. The call for participation of healthy volunteers was executed by advertisements on social media, flyers, and online local news sites in the Netherlands. The healthy volunteers were asked to complete the online questionnaire containing demographic variables and the CSI part A and B via REDCap [35]. Exclusion criteria for the healthy volunteers were the experience of any current pain or pain in the previous week, the use of pain medication or undergoing pain treatment, being diagnosed with a CSS (based on the CSI part B) or reporting the use of anti-depressant or anti-epileptic drugs at the moment of completing the questionnaire. 

### 2.2. Measures

Participating patients’ and healthy volunteers’ descriptives consist of sex, age, weight, height, BMI and CSI total score. The patient descriptives also include pain severity, pain catastrophizing, pain disability, number of pain locations, number of reported CSSs, health-related quality of life, pain location(s), and pain disorders. 

#### 2.2.1. Central Sensitization Inventory (CSI)

The CSI [6,8] consists of two parts. The CSI is a screening tool for the possible presence of CSS. It is intended be used as an indication for the possible presence of HACS. Part A assesses 25 health-related items supposed to be associated with HACS. Each item is measured on a 5-point Likert scale ranging from 0 (never) to 4 (always), and the total score ranges from 0 to 100 with a previously established cutoff value of 40 or higher to address the possible presence of HACS. The original CSI [6], and the Dlv [8] showed good internal consistency and test–retest reliability. Part B evaluates the presence of seven CS-associated disorders, i.e., tension headaches/migraines, fibromyalgia, irritable bowel syndrome, restless leg syndrome, temporomandibular joint disorder, chronic fatigue syndrome, and multiple chemical sensitivities; and three associated diagnoses, i.e., depression, anxiety/panic attacks, and neck injury [6]. The number of reported CSSs is based on the presence of one or more CSSs in the CSI part B.

#### 2.2.2. Pain Catastrophizing Scale (PCS)

The PCS [36] assesses thoughts, feelings, and cognitions about pain. For each of the 13 statements, the participant is asked to answer on a scale from 0 (totally not) to 4 (always), and the total score ranges from 0 to 52. The higher the score, the more catastrophizing thoughts are present. The original PCS (Cronbach’s alpha = 0.93 [37], test–retest reliability = 0.75 [36]), as well as the PCS-Dlv (Cronbach’s alpha between 0.85 and 0.9138–40, test–retest reliability = 0.7341), show good reliability.

#### 2.2.3. Pain Disability Index (PDI)

The PDI [38,39] is used to measure how aspects of the patient’s life are disrupted by chronic pain. It lists seven life activity categories (family/home responsibilities, recreation, social activity, occupation, sexual behavior, self-care, and life-support activities), which have to be scored by the patient between 0 (no limitations) to 10 (totally limited). The total score ranges from 0 to 70. A higher score reflects a higher interference of pain with daily activities. The original PDI shows good internal consistency (Cronbach’s alpha = 0.87) [40], and the test–retest reliability is good (ICC = 0.91) [41]. The PDI-Dlv has good internal consistency (Cronbach’s alpha = 0.85) [39] and sufficient test–retest reliability (ICC = 0.78) [39] in patients with chronic low back pain.

#### 2.2.4. SF-12 Health Survey (SF-12)

The SF-12 [42,43] is a shorter version of the SF-36, a participant-reported health-related quality of life survey. It consists of eight sections: vitality, physical functioning, bodily pain, general health perceptions, physical role functioning, emotional role functioning, social role functioning, and mental health. The total score ranges from 0 to 100, and a higher score indicates a better health status. The SF-12 showed sufficient and good internal consistency with Cronbach’s alpha 0.77 [44] (Physical subscale) and Cronbach’s alpha 0.80 [44] (Mental subscale) in patients with back pain. Test–retest reliability shows to be sufficient with ICC = 0.79 [45].

#### 2.2.5. Numeric Rating Scale (NRS) for Pain

The NRS is a pain rating scale in which patients and healthy volunteers were asked to rate their current pain, mean pain, and worst pain over seven days (0 = no pain; 10 = maximum pain imaginable). The NRS shows good test–retest reliability (ICC = 0.95) [46]. The average pain score for the past seven days was used to assess the factor pain severity.

#### 2.2.6. Pain Location

Patients’ pain location was based on self-reported questionnaires used in the pain center. The categories that could be chosen were “head”, “neck, shoulders, high back, and arms”, “elbow, wrist, hand”, “lower back”, “hip, knee”, “foot, ankle”, “chest, abdomen”, “pelvic” and “other”. When the pain was present in more than one location, these patients were categorized as having pain in “multiple locations”. The number of pain locations was based on the number of chosen categories.

#### 2.2.7. Pain Disorder

Pain disorder was categorized in ICD-11 codes [47]: chronic primary pain, chronic cancer pain, chronic posttraumatic and post-surgical pain, chronic neuropathic pain, chronic headache and orofacial pain, chronic visceral pain, and chronic musculoskeletal pain [48,49]. Because ICD-10 codes were used in hospital administration systems when patients were included, these were extracted from the patient files and converted to ICD-11 codes using the “10To11MapToOneCategory” file from the World Health Organization [47,50].

### 2.3. Sample Size Calculation for the Number of Healthy Volunteers

For the healthy volunteers, a sample size table was used [51] to calculate the required sample size. With a confidence level of 95%, a margin of error of 5.0%, and a population size of 10 million people >18 years old, 384 healthy volunteers were needed.

### 2.4. Data Analysis

Patients were divided into two groups: patients with one or more CSSs (CSS+) and patients without a CSS (CSS−) based on the CSI part B. Descriptive data are presented per group (total group and per sex separately), CSS+/CSS−, and healthy volunteers. Mean ± standard deviation (SD) was applied when the data were normally distributed and a median and interquartile range [IQR] when not normally distributed (examined via Kolmogorov–Smirnov test). Nominal data are shown in the presence with percentages (n (%)). 

For the analyses of our primary aim, the establishment of the cutoff points for the CSI, ROC analyses [52] was conducted. The presence of CSS in patients, defined as at least one CSS based on the CSI part B, versus healthy volunteers without CSS [10] was used in the ROC analyses. The area under the curve (AUC), sensitivity, specificity, positive and negative predictive values, and positive and negative likelihood ratios were calculated [53,54]. The optimal cutoff point was determined based on the Youden index [55] for CSS+ vs. healthy volunteers. This index is defined as J = max_c_ (Se(c) + SP(c) − 1) and ranges from 0 to 1, with 1 being the optimal diagnostic accuracy. J provides a criterion for choosing the “optimal” threshold value (c*), the threshold value for which Se(c) + Sp(c) − 1 is maximized (max_c_) [55,56].

For the analyses of the second aims, differences between the groups (patients compared to healthy volunteers, comparison between males and females, and the comparison between patients with or without a CSS) were tested using the independent student’s *t*-test in normally distributed data and the Wilcoxon signed-rank test when not normally distributed. For the location of pain, the Χ^2^-test was used to calculate the differences between CSS+ and CSS−. The number of CSS and the type(s) of CSS were assessed with percentages.

Univariate regression and multivariate regression analyses were used to identify possible predictors for the CSI score and to analyze the associations between independent factors (sex, age, number of central sensitivity syndromes, number of pain locations, pain catastrophizing, pain disability, and pain severity) and CSI score. To address factors associated with the CSI score, a multiple linear regression analysis (backwards method) was performed. With ten factors (sex, age, BMI, pain severity, pain catastrophizing, pain disability, number of pain locations, number of reported CSSs, physical component and mental component SF-12), there are enough patients included in this study (>10 cases per factor [57]) for regression analyses. To choose the selection parameter to decide whether an effect should be retained in the model, 0.01 was used for inclusion and 0.05 for exclusion. This is because of a participant per factor ratio of 100 or above and is based on the Akaike information criterion prognostic models [58,59]. When the selection was made, the included factors were used in linear regression analyses (method enter) to create a regression equation with all the predictor variables. The same analysis is also performed for both sexes separately. 

The data and metadata will be stored at the repository at the UMCG, ensuring the data’s security and backup. UMCG pursues a FAIR data policy for research conducted in the UMCG. To make the data findable and accessible to others, we included a description in the UMCG data catalog data: https://www.groningendatacatalogus.nl, accessed on 19 July 2023. The data and metadata are available for researchers inside and outside the institute. This catalog is in sync with relevant (inter)national catalogues, such as Biobanking and Biomolecular Resources Research Infrastructure and National Academic Research and Collaborations Information System. A data access committee has been put in place to review requests and assure the accessibility of the data. This access committee can be reached via the corresponding author.

## 3. Results

Patient inclusion resulted in 2890 patients consisting of 1813 women (62.7%) (1213 women CSS+ (66.9%) and 600 women CSS− (33.1%)) and 1077 men (37.3%) (517 men CSS+ (48.0%) and 560 men CSS− (52.0%)) with chronic pain (Figure 1a). During patient inclusion, 1036 measurements were excluded because CSI part B was not completed by the patient or was not asked for. The healthy volunteers consisted of 157 women (62.8%) and 93 men (37.2%), a total of 250 participants (Figure 1b). Descriptives are presented for the total group of patients (Table 1) and healthy volunteers (Table 2) and are separated between women and men, and CSS+ and CSS−. The patient’s type of pain, based on the ICD-11 of all participating patients, are presented in Figure 2.

Tree diagram of the ICD-11 codes of all reported ICD-11 codes used 1.0% or more. Some patients reported more than 1 ICD-11 code, but only the main complaint is used. N: number; ♀: women; ♂: men. The total number of patients: n = 2890. Excluded from the tree diagram: codes with a prevalence of <1.0% (n = 511 (17.7%)) and not specified (n = 69 (2.4%)).

### 3.1. Cutoff Scores for the CSI

Cutoff scores for the CSI were established by comparing CSS+ with healthy volunteers. The ROC analysis resulted in an AUC of 0.95 with the highest Youden index (0.78) at a cutoff score of 30 points on the CSI, resulting in a sensitivity of 85% and a specificity of 93%. (Table 3 and Appendix A). For women, the highest Youden index (0.79) resulted in an AUC of 0.96 and a cutoff score of 33 with a sensitivity of 83% and a specificity of 97% (Table 3). In men, the AUC was 0.95, and the highest Youden index was 0.80, resulting in a cutoff score of 25 with a sensitivity of 89% and a specificity of 91% (Table 3). The total group of CSS+ and healthy volunteers resulted in a positive predictive value of 98.8%, a negative predictive value of 47.8%, a positive likelihood ratio of 11.9, and a negative likelihood ratio of 0.16. In women, the analysis resulted in a positive predictive value of 99.5%, a negative predictive value of 41.8%, a positive likelihood ratio of 25.9, and a negative likelihood ratio of 0.18. In men, it was 98.3%, 59.9%, 10.34, and 0.12, respectively.

### 3.2. Differences between Groups

Significant differences are observed for the CSI score between CSS+ and CSS− and also when examining the sexes separately (Table 1). When comparing the CSS+ and CSS− with healthy volunteers, significant differences are found in almost all descriptives, except for height, when men with a CSS are compared with healthy male volunteers, and the total group of healthy volunteers is compared with all patients without a CSS (Table 2).

Of the 2890 patients, 59.9% (n = 1730) were diagnosed with one or more CSSs based on the CSI part B. The number of CSSs in the patient group ranges from 0 to 8. Women reported more often one or more CSSs (66.9%, 1213 out of 1813) than men (48.0%, 517 out of 1077). With the increase in the number of CSSs, the mean CSI score increases except for the small number of patients with 8 CSSs. The three most frequently diagnosed CSSs were depression (23.4%), migraine or tension headaches (18.7%), and irritable bowel syndrome (16.5%). In women, the three most frequent CSSs diagnosed were depression (25.4%), migraine or tension headaches (23.3%), and fibromyalgia (21.8%). In men, it was depression (20.1%), followed by neck injury (including whiplash) (12.2%), and migraine or tension headaches (11.0%). 

### 3.3. Associated Factors with the CSI Score

Univariate regression showed that all studied factors were significantly associated with the CSI score, for the total patient group, as well as both sexes separately (Table 4).

For the total patient group, multiple backward linear regression analyses showed that all studied factors, i.e., sex, age, BMI, pain severity, PCS score, PDI score, number of pain locations, number of reported CSSs, the physical and mental component of the SF-12, were associated factors for the CSI score (Appendix A). For women, all studied factors (as mentioned for the total patient group), excluding sex and pain severity, were associated with the CSI score (Appendix A). In men, the associated factors for the CSI score were the same as in women, except for age and BMI (Appendix A). A model was created to predict the CSI score with the associated factors, as shown in Table 5. Based on this model, formulas can be created by multiplying factors with the unstandardized beta. 

The formula based on the prediction model is for all patients: 69.71 + sex*2.79 − age*0. 07+ BMI*0.10 − pain severity*0.36 + pain catastrophizing*0.16 + pain disability*0.13 + number of pain locations*2.18 − number of reported CSSs*3.24 – physical component SF-12*0.40 − mental component SF-12*0.73.

For women the formula is 79.43 − age*0.10 + BMI*0.13 + pain catastrophizing*0.09 + pain disability*0.10 + number of pain locations*2.24 + number of CSSs*3.21 − physical component SF-12*0.40 − mental component SF-12*0.80.

In men, the formula is 65.38 + pain catastrophizing*0.23 + pain disability*0.12 + number of pain locations*2.05 + the number of reported CSSs*3.23 − physical component SF-12*0.44 − mental component SF-12*0.63. The number of reported CSSs is the most associated factor, followed by sex and the number of pain locations.

## 4. Discussion

This study resulted in a CSI cutoff value of 30 in the total group, 33 for women, and 25 for men comparing patients with CSS(s) to healthy volunteers. 

The secondary aims of this study revealed high discrimination between the CSS+ group and healthy volunteers. Psychometric analyses in the total group and for sexes separately showed a good probability of a positive test in patients with HACS and a negative test in patients without HACS. The analysis of the CSI score showed that all factors were associated with the CSI score for the total group. The associated factors for the CSI score were sex, age, BMI, pain severity, pain catastrophizing, pain disability, number of pain locations, number of reported CSSs and quality of life. In women, pain severity was eliminated as an associated factor. In men, age, BMI and pain severity were eliminated. The relatively low adjusted R^2^, thus the explained variance, in this study suggest that more and/or other factors might contribute to the CSI score such as pain sensitivity [23,25,34,60,61] and heart rate variability [23,25,34,60,61,62].

In this study, a larger sample of patients and healthy volunteers with a similar age were used compared to previous studies [10,32]. Neblett and colleagues [10] determined the presence of HACS using a list of eleven CSSs based on the seven CSSs stated in the CSI part B but without the three CS-associated syndromes. However, they added four extra diagnoses: myofascial pain syndrome, posttraumatic stress disorder, interstitial cystitis, and complex regional pain syndrome [10]. This list differs from the CSSs from the CSI part B [12] which was used in our and other studies [11,14,32,62,63] and thus the extra disorders with a HACS component were not structurally examined. 

In the original study reporting the development and the validity of the CSI by Neblett and colleagues, the cutoff value was based on the specificity of at least 75% combined with the highest sensitivity score [10,32]. In our and other studies [11,13,32], the Youden index was used. The Youden index reflects the intention of maximizing overall correct classification rates and thus minimizing misclassification rates [53,64]. Using the same method as Neblett and colleagues in our study, our cutoff values would be lower than established now, with higher sensitivity scores but lower specificity scores. 

In our and previous studies [10,12,63] the presence of at least one CSS was indicative of the presence of HACS to establish cutoff values. In other studies, the presence of three or more CSSs [14] or the presence of musculoskeletal disorders [32] was indicative of the presence of HACS. We used the criteria of at least one CSS because previous research established a strong correlation between the CSI and HACS diagnoses [10]. The presence of a CSS indicates that the central nervous system is involved in this patients’ pain. Furthermore, there are also studies in which QST measures [11,25,65] were found helpful to measure physical sensitivity as an indicator of the presence of HACS. Finally, there is also a study where the presence of pain was used to establish a cutoff value for the CSI [13]. However, in our second analysis, we showed that the association between pain severity and the CSI score is not significant when analyzing the sexes separately, supporting the idea that presence of HACS is not the same as the presence of pain.

In the paper by Neblett and colleagues [10], the cutoff score was based on a comparison with healthy volunteers and resulted in a score of 40 or higher. Another study used a comparison with healthy volunteers to establish a cutoff value of 37 [32]. Most studies established cutoff scores of the CSI without a comparison with the healthy population as the control [11,12,13,14,25,63,65] resulting in, for example, a cutoff score ranging from 17 [11]–36 [25] in patients with knee osteoarthritis. A comparison between CSI outcomes from a healthy population and an individual patient helps to interpret a patient’s outcome [66,67]. This is because it might indicate the severity of HACS or the reduction in the difference between patients and the healthy population. Moreover, it might show treatment success. The most optimal CSI cutoff value is supposed to help discriminate more reliably between patients suffering from chronic pain with HACS and healthy volunteers, and to distinguish between patients suffering from chronic pain with and without HACS. 

In previous studies addressing the CSI, uniformity lacks in factors that were correlated with the CSI score. Some studies presented correlated factors [15,23,24,27,31] (sex, beliefs about exercise and pain, pain catastrophizing, pain severity, and widespread pain), but in other studies, these factors were not associated with the CSI score [15,23,24]. For example, in our study, pain catastrophizing is a predictor for the CSI score, which is in line with some studies showing a good correlation [23,24] between the pain catastrophizing scale and the CSI. However, other studies showed a moderate [25,26,27] or low [15,28] correlation.

The second analysis in this study shows that pain severity is not associated with the CSI score when women and men are assessed separately. This suggests that pain intensity does not influence the CSI score and vice versa. The association between pain severity and the CSI score in the total patient group is statistically significant, but this might be caused by the differences between sexes that are canceled out when analyzed as a group.

Importantly, there was a sex difference in the established cutoff values for the CSI score and the associated factors confirming previous research [22,23]. The cutoff value for women is higher than for men. Differences in pain perception between sexes align with other studies [18,19,21,68]. Women perceive more clinical pain disorders [18], more pain severity [18,20], lower pain thresholds [19], greater ability to discriminate different kinds of pain [19], less tolerance of noxious stimuli [19], and experience more visceral pain [21] compared to men. These differences are also shown in immune system cell populations [20].

The CSI-Dlv is used in the Netherlands [8,15] and the Dutch-speaking part of Belgium [8,9]. In our study, the mean for the CSI was 40.2. This is within the range of other studies using the CSI-Dlv, ranging from 36.1 [15] to 43.9 [9] in patients with chronic pain. In the study by Kregel and colleagues [8], the mean CSI score of the healthy volunteers was 21.6 compared to 16.4 in our study. This might be explained by our stricter sample criteria for ensuring our healthy volunteers being indeed healthy. Their exclusion criteria were no chronic pain or long-term pain during the past five years. In our study, participants using pain medication, anti-epileptics, and anti-depressives who experienced pain during the past week and were currently undergoing pain treatment were excluded. 

### 4.1. Limitations and Strengths

To have a good representation of the actual population of patients with chronic pain, we included 2890 patients with pain from different origins. We used strict selection criteria for the healthy volunteers, which resulted in 250 instead of the necessary 384 healthy volunteers. 

To establish cutoff values for the CSI-Dlv, we included only Dutch-speaking participants. Country and culture might impact the CSI score [10,69], as shown in a multi-country study [70], and from comparing results of studies performed in different countries [71,72]. Therefore, we have to be careful to extrapolate these results to other languages and countries.

In future research, the inclusion of psychophysical tests such as QST might be recommended to address the presence of HACS in patients besides the use of questionnaires [11,25,34,60,65,73]. A recent systematic review showed a weak or no correlation between the CSI and QST and found a strong correlation between the CSI and psychological measures [68]. A weak or no correlation does not mean that these indicators cannot be combined. The combination of questionnaires and QST might help to address patients with HACS according to a recently proposed HACS grading system [5]. In addition, a recent review also recommended that a combination of features and methods can be used to discriminate between mechanism-based categories of pain [74]. Questionnaires such as Multisensory Amplification Scale might also give some insights into a related construct of generalized central sensitivity and/or risk factor for HACS [73].

Using a cutoff score in the CSI creates an interpretation favoring dichotomous outcomes. A categorical distribution [7,75,76,77] or a continuous scale may give more clinical insights, suggesting that HACS is present to a greater or lesser extent, which might be a better interpretation of a biological phenomenon. In a recent systematic review, it was concluded that a higher CSI score suggests a higher chance of HACS being present, but it does not reflect the increased nociceptive response [68].

Future research may determine if a categorical or continuous distribution is more suitable than a dichotomous outcome and how these severity levels may be established.

### 4.2. Clinical Implications

The newly proposed cutoff values will better distinguish between patients with and without HACS because of higher sensitivity and specificity that might result in a higher prevalence of HACS. However, the positive and negative predictive values should also be considered. A patient should be evaluated on more indicators for HACS, such as temporal summation and conditioned pain modulation [5], to determine if HACS is present. Especially for patients with a CSI score below the cutoff value because of the low negative predictive value, it may be a false negative.

## 5. Conclusions

In conclusion, new cutoff values for the CSI are established in our study, being 33 for women, 25 for men, and 30 out of 100 for both sexes together. This study revealed high discrimination between the CSS+ group and healthy volunteers. The analysis of the CSI score showed that all factors (sex, age, BMI, pain severity, pain catastrophizing, pain disability, number of pain locations, number of reported CSSs, and quality of life) were associated with the CSI score for the total group. When sexes were analyzed separately, pain severity was not an associated factor. Sex is an important associated factor for the CSI score and should be taken into account in the assessment of HACS via the CSI.

## Figures and Tables

**Figure 1 jcm-12-04849-f001:**
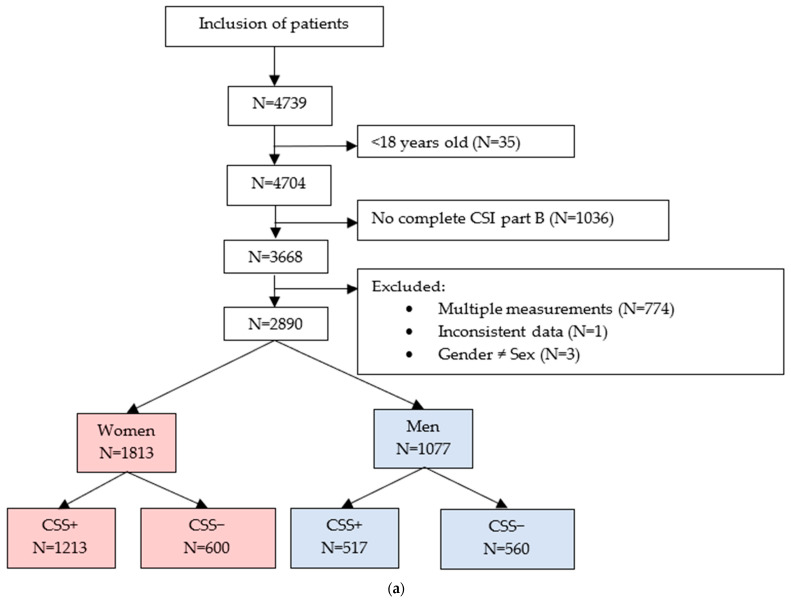
(**a**). Flowchart showing the inclusion of patients. (**b**). Flow chart showing the inclusion of healthy volunteers. Abbreviations: N: number; CSI: central sensitization inventory; CSS: central sensitivity syndromes.

**Figure 2 jcm-12-04849-f002:**
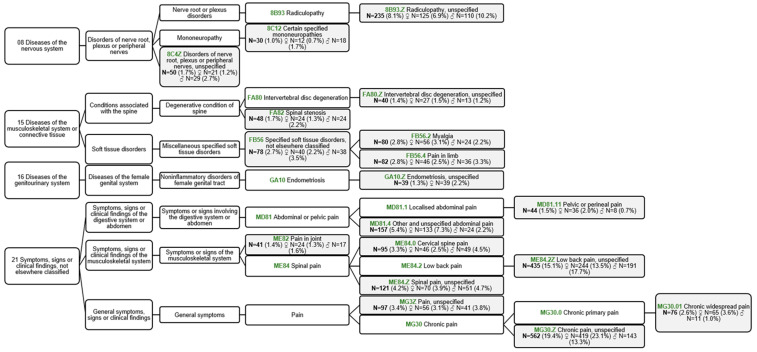
Tree diagram of patients’ type of pain based on ICD-11; Green: ICD-11 code.

**Table 1 jcm-12-04849-t001:** Descriptives of the patients.

Patients	Total Group					Women					Men						CSS+		CSS−
	CSS+		CSS−			CSS+		CSS−			CSS+		CSS−			Women vs. Men		Women vs. Men	
	N	Mean ± SD %	N	Mean ± SD %	*p*	N	Mean ± SD %	N	Mean ± SD %	*p*	N	Mean ± SD %	N	Mean ± SD %	*p*	*p*		*p*	
Age (years)	1730	49.5 ± 15.2	1160	50.8 ± 17.1	0.028 *	1213	47.9 ± 15.2	600	47.9 ± 17.9	0.947	517	53.1 ± 14.5	560	54.1 ± 15.7	0.326	<0.001 *		<0.001 *	
Height (cm)	1370	173.2 ± 9.5	992	175.2 ± 10.1	<0.001 *	955	169.2 ± 7.2	508	168.7 ± 7.2	0.287	415	182.6 ± 7.1	484	182.0 ± 8.0	0.210	<0.001 *		<0.001 *	
Weight (kg)	1370	82.4 ± 18.5	992	82.2 ± 17.9	0.882	955	78.3 ± 17.1	508	74.8 ± 16.2	<0.001 *	415	91.8 ± 18.3	484	90.0 ± 16.3	0.135	<0.001 *		<0.001 *	
BMI (kg/cm²)	1369	27.4 ± 5.5	992	26.7 ± 5.1	0.002 *	954	27.4 ± 5.8	508	26.3 ± 5.7	0.001 *	415	27.5 ± 5.0	484	27.1 ± 4.3	0.286	0.717		0.008 *	
CSI (0−100)	1730	44.8 ± 14.5	1160	33.3 ± 14.2	<0.001 *	1213	46.0 ± 14.4	600	34.6 ± 14.4	<0.001 *	517	42.0 ± 14.3	560	31.9 ± 13.9	<0.001 *	<0.001 *		0.001 *	
Pain now (0−10)	1147	6.3 ± 2.2	919	6.1 ± 2.3	0.026 *	785	6.3 ± 2.2	463	6.0 ± 2.3	0.060	362	6.3 ± 2.2	456	6.1 ± 2.3	0.198	0.853		0.651	
Mean pain last 7 days (0–10)	1430	6.7 ± 1.8	1024	6.7 ± 1.8	0.680	991	6.7 ± 1.8	523	6.6 ± 1.8	0.382	439	6.7 ± 1.8	501	6.7 ± 1.8	0.826	0.958		0.296	
Worst pain (0–10)	1430	8.5 ± 1.4	1024	8.4 ± 1.5	0.155	991	8.5 ± 1.3	523	8.4 ± 1.5	0.085	439	8.4 ± 1.5	501	8.4 ± 1.4	0.978	0.237		0.667	
PCS (0–52)	1686	22.1 ± 11.6	1145	21.8 ± 11.4	0.445	1171	21.3 ± 11.3	586	20.9 ± 11.4	0.533	515	24.0 ± 11.9	559	22.7 ± 11.3	0.059	<0.001 *		0.009 *	
PDI (0–70)	1170	40.4 ± 13.0	800	38.4 ± 14.7	0.001 *	797	40.4 ± 12.9	388	37.9 ± 15.0	0.005 *	373	40.5 ± 13.3	412	38.9 ± 14.4	0.092	0.878		0.349	
Number of pain locations	1730	2.5 ± 2.1	1160	1.8 ± 1.3	<0.001 *	1213	2.6 ± 2.2	600	1.8 ± 1.3	<0.001 *	517	2.3 ± 1.9	560	1.7 ± 1.3	<0.001 *	0.001 *		0.159	
Number of CSSs (0–10)	1730	2.0 ± 1.2	1160	0	<0.001 *	1213	2.1 ± 1.2	600	0	<0.001 *	517	1.7 ± 1.0	560	0	<0.001 *	<0.001 *		NA	
Physical component SF-12 (0–100)	1582	38.9 ± 5.6	1090	39.3 ± 5.9	0.090	1104	38.7 ± 5.5	566	39.0 ± 5.7	0.303	478	39.3 ± 5.9	524	39.6 ± 6.1	0.463	0.085		0.154	
Mental component SF-12 (0–100)	1582	45.2 ± 7.3	1090	47.4 ± 7.3	<0.001 *	1104	45.4 ± 7.1	566	47.5 ± 7.3	<0.001 *	478	44.6 ± 7.8	524	47.2 ± 7.4	<0.001 *	0.060		0.615	
Pain locations					<0.001 *					<0.001 *					0.003 *	0.008 *		<0.001 *	
Head	35	2.0%	24	2.1%		23	1.9%	12	2.0%		12	2.3%	12	2.1%					
Neck, shoulders, high back and arms	102	5.9%	66	5.7%		59	4.9%	24	4.0%		43	8.3%	42	7.5%					
Elbow, wrist, hand	14	0.8%	6	0.5%		8	0.7%	3	0.5%		6	1.2%	3	0.5%					
Lower back	323	18.7%	315	27.2%		213	17.6%	155	25.8%		110	21.3%	160	28.6%					
Hip, knee	30	1.7%	20	1.7%		19	1.6%	9	1.5%		11	2.1%	11	2.0%					
Foot, ankle	24	1.4%	40	3.4%		16	1.3%	12	2.0%		8	1.5%	28	5.0%					
Chest, abdomen	93	5.4%	72	6.2%		72	5.9%	47	7.8%		21	4.1%	25	4.5%					
Pelvis	61	3.5%	39	3.4%		49	4.0%	29	4.8%		12	2.3%	10	1.8%					
Other	121	7.0%	104	9.0%		79	6.5%	49	8.2%		42	8.1%	55	9.8%					
Multiple locations	927	53.6%	474	40.9%		675	55.6%	260	43.3%		252	48.7%	214	38.2%					

Abbreviations: CSS+: patients with chronic pain with central sensitivity syndrome(s); CSS−: patients with chronic pain without central sensitivity syndrome(s); N: number; BMI: body mass index; PCS: pain catastrophizing scale; PDI: pain disability index; CSI: central sensitisation inventory; CSS: central sensitivity syndrome(s); NA: Not applicable; SF-12: short-form 12-item health questionnaire; SD: standard deviation. Statistics: Comparison made with a student *t*-test between women vs. men and CSS+ vs. CSS−. For the location of pain, Χ^2^-test was used to calculate the differences between groups. * *p* < 0.05.

**Table 2 jcm-12-04849-t002:** Descriptives of the healthy volunteers.

Healthy Volunteers	Total Group		Women		Men		Women vs. Men	CSS+ vs. Healthy Volunteers			CSS− vs. Healthy Volunteers		
								Total group	Women	Men	Total group	Women	Men
	N	Mean ± SD	N	Mean ± SD	N	Mean ± SD	*p*	*p*	*p*	*p*	*p*	*p*	*p*
**Age (years)**	250	43.2 ± 16.1	157	41.6 ± 14.2	93	45.9 ± 18.6	0.056	<0.001 *	<0.001 *	<0.001 *	<0.001 *	<0.001 *	<0.001 *
**Height (cm)**	250	175.8 ± 9.2	157	170.9 ± 6.2	93	184.0 ± 7.2	<0.001 *	<0.001 *	0.004 *	0.093	0.412	0.001 *	0.025 *
**Weight (kg)**	250	75.2 ± 14.9	157	69.9 ± 12.7	93	84.2 ± 14.1	<0.001 *	<0.001 *	<0.001 *	<0.001 *	<0.001 *	<0.001 *	0.001 *
**BMI (kg/cm²)**	250	24.2 ± 3.8	157	23.9 ± 4.0	93	24.8 ± 3.3	0.067	<0.001 *	<0.001 *	<0.001 *	<0.001 *	<0.001 *	<0.001 *
**CSI (0–100)**	250	16.4 ± 8.6	157	17.3 ± 8.7	93	15.1 ± 8.3	0.049 *	<0.001*	<0.001 *	<0.001 *	<0.001 *	<0.001 *	<0.001 *

Abbreviations: CSS+: patients with chronic pain with central sensitivity syndrome(s); CSS−: patients with chronic pain without central sensitivity syndrome(s); N: number; BMI: body mass index; CSI: central sensitization inventory; SD: standard deviation. Statistics: Comparison made with an unpaired student *t*-test between women vs. men and CSS+ vs. CSS−. For the location of pain Χ^2^-test was used to calculate the differences between groups. * *p* < 0.05.

**Table 3 jcm-12-04849-t003:** Assessment of the CSI cutoff scores for the total group and per sex, comparing patients with CSS+ and healthy volunteers.

Total Group (N = 1730) vs. Healthy Volunteers (N = 250); AUC: 0.953			
Cutoff Score	Youden Index	Sensitivity	Specificity
28	0.7775	0.8855	0.8920
29	0.7785	0.8705	0.9080
30	0.7818	0.8538	0.9280
31	0.7701	0.8341	0.9360
32	0.7648	0.8208	0.9440
Women (N = 1213) vs. Healthy Female Volunteers (N = 157) AUC: 0.956			
Cutoff Score	Youden Index	Sensitivity	Specificity
30	0.7913	0.8805	0.9108
31	0.7851	0.8615	0.9236
32	0.7782	0.8483	0.9299
33	0.7934	0.8252	0.9682
34	0.7736	0.8054	0.9682
35	0.7505	0.7824	0.9682
Men (N = 517) vs. Healthy Male Volunteers (N = 93) AUC: 0.947			
Cutoff Score	Youden Index	Sensitivity	Specificity
23	0.7497	0.9110	0.8387
24	0.7674	0.9072	0.8602
25	0.8037	0.8897	0.9140
26	0.7883	0.8743	0.9140
27	0.7708	0.8569	0.9140

Abbreviations: N: number; CSI: central sensitization inventory; CSS: central sensitivity syndromes; AUC: area under the curve. Statistics: ROC analyses were used to calculate AUC, sensitivity and specificity, and the Youden index.

**Table 4 jcm-12-04849-t004:** Univariate regression analyses for the total patient group and per sex in relation to the CSI total score.

	Total Group						Women						Men					
	N	R	adj. R^2^	I	β	*p*	N	R	adj. R^2^	I	β	*p*	N	R	adj. R^2^	I	β	*p*
Sex	2890	0.171	0.029	31.314	5.450	<0.001												
Age	2890	0.158	0.025	47.803	−0.152	<0.001	1813	0.133	0.017	48.266	−0.126	<0.001	1077	0.132	0.016	43.746	−0.130	<0.001
BMI	2361	0.088	0.007	32.450	0.251	<0.001	1462	0.120	0.014	32.834	0.318	<0.001	899	0.043	0.001	32.085	0.134	0.201
Pain severity	2454	0.212	0.045	27.321	1.784	<0.001	1514	0.192	0.036	30.609	1.608	<0.001	940	0.260	0.067	21.574	2.134	<0.001
PCS	2831	0.328	0.107	30.512	0.440	<0.001	1757	0.318	0.101	33.209	0.429	<0.001	1074	0.406	0.164	24.570	0.521	<0.001
PDI	1970	0.401	0.160	22.876	0.449	<0.001	1185	0.383	0.146	25.989	0.428	<0.001	785	0.446	0.198	18.161	0.481	<0.001
Number of Pain locations	2890	0.400	0.160	32.730	3.344	<0.001	1813	0.410	0.168	34.580	3.225	<0.001	1077	0.353	0.125	30.255	3.262	<0.001
Number of CSSs	2890	0.467	0.218	33.764	5.332	<0.001	1813	0.459	0.210	35.160	4.935	<0.001	1077	0.429	0.183	32.027	5.745	<0.001
PC SF-12	2672	0.215	0.046	62.279	−0.574	<0.001	1670	0.205	0.041	63.320	−0.558	<0.001	1002	0.219	0.047	57.835	−0.545	<0.001
MC SF-12	2672	0.518	0.268	89.414	−1.076	<0.001	1670	0.515	0.264	91.992	−1.086	<0.001	1002	0.549	0.301	85.461	−1.068	<0.001

Abbreviations: N: Number of patients; R: Correlation; adj. R^2^: adjusted determination coefficient (adjusted R squared); I: Intercept; β: unstandardized regression coefficient; *p*: *p*-value; BMI: body mass index; PCS: pain catastrophizing scale; PDI: pain disability index; CSSs: central sensitivity syndromes; PC SF12: physical component outcome of the SF-12 health survey; MC SF12: mental component outcome of the SF-12 health survey. Statistics: Simple, univariate, linear regression; statistically significant *p*-value < 0.05.

**Table 5 jcm-12-04849-t005:** Models to predict the CSI score for the total group and per sex.

Group	Total Group		Women		Men	
unadjusted R^2^	0.580		0.579		0.550	
adjusted R^2^	0.577		0.576		0.545	
	β	CI 95%	β	CI 95%	β	CI 95%
Constant	69.71	60.88–78.53	79.43	68.35–90.52	65.38	53.67–77.08
Sex	2.79	1.72–3.86	-	-	-	-
Age(years)	−0.07	−0.10–−0.03	−0.10	−0.15–−0.05	-	-
BMI	0.10	0.01–0.20	0.13	0.01–0.24	-	-
Pain severity (NRS)	−0.36	−0.72–−0.01	-	-	-	-
Pain catastrophizing (PCS)	0.16	0.11–0.21	0.09	0.02–0.16	0.23	0.16–0.31
Pain disability (PDI)	0.13	0.08–0.18	0.10	0.04–0.16	0.12	0.05–0.19
Number of pain locations	2.18	1.91–2.44	2.24	1.91–2.56	2.05	1.59–2.50
Number of CSSs (CSI part B)	3.24	2.83–3.66	3.21	2.71–3.71	3.23	2.50–3.96
Physical component (SF-12)	−0.40	−0.50–−0.30	−0.40	−0.54–−0.25	−0.44	−0.60–−0.29
Mental component (SF-12)	−0.73	−0.82–−0.64	−0.80	−0.92–−0.67	−0.63	−0.76–−0.50

Abbreviations: β: unstandardized beta; CI 95%: confidence interval of 95%; BMI: body mass index; NRS: numeric rating scale for pain; PCS: pain catastrophizing scale; PDI: pain disability index; CSSs: central sensitivity syndromes; CSI: central sensitization inventory; SF12: short-form 12-item health questionnaire. Statistics: Multiple regression analysis (backwards).

## Data Availability

The data and metadata will be stored at the repository at the UMCG, ensuring the data’s security and backup. UMCG pursues a FAIR data policy for research conducted in the UMCG. To make the data findable and accessible to others, we included a description in the UMCG data catalog data: https://www.groningendatacatalogus.nl (accessed on 19 July 2023). The data and metadata are available for researchers inside and outside the institute. This catalog is in sync with relevant (inter)national catalogues, such as Biobanking and Biomolecular Resources Research Infrastructure and National Academic Research and Collaborations Information System. A data access committee has been put in place to review requests and assure the accessibility of the data. This access committee can be reached via the corresponding author.

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
