# Peer review of "Validity of the Central Sensitization Inventory to Address Human Assumed Central Sensitization: Newly Proposed Clinically Relevant Values and Associations"

_jcm, 2023, doi:10.3390/jcm12144849_

Round 1

Reviewer 1 Report

An important topic and the conclusions.  I found it a bit long and incredibly heavy with tables that were hard to follow; i think that the paper would have more effect if it could be streamlined to bit.  I am not that great with stats, but the fact that a CSI inventory floor needs to lowered is an important conclusion

fine

Reviewer 2 Report

Overall the authors have done a comprehensive and thorough work. I have, however,  some comments as follows:

TITLE

I consider the term “established” used in the title as too strong. Establish gives the idea of something definite, on which has been agreed on or is already settled, whereas to my understanding the cut-off values shown in the present article, are different from previous ones but have not been in any way settled by a large community yet.

Therefore, I’d suggest to change to “proposed” or to tone down somehow.

INTRODUCTION

-Lines 36-39

“The use of the term CS in clinical practice……phathophysiological mechanism.”

The affirmation in the first sentence is a bit strong, considering that the authors are citing in citation 2 an article which gives a description of “Central sensitization in human volunteers”. I would, therefore, suggest to change the two phrases in something like this:

The use of the term CS in clinical practice has been under debate mainly because it is difficult to  to distinguish between patients with and without CS based on objective pathophysiological mechanisms.

-Also, in the following phrase I’d suggest to modify it by removing “therefore, we suggested”, since the cited work is not part of this manuscript. I’d suggest changing  to “It has been previously suggested”.

-Similarly, phrase line 58The number of central Sensitization syndromes (CSSs) was different between men and women with 59 chronic musculoskeletal pain disorders.[22]” . I’d suggest changing to “has been reported to be different”

-In line 68 page 2, where the main aim is stated, I suggest to change it by summarizing briefly the aim without including details on the methodology.

METHODS

-Page 3 lines 110-113

I don’t think I understand very well what the authors meant by the following phrase: “The original  CSI (Cronbach’s alpha = 0.88, test-retest reliability = 0.82)[6], as well as the in this study, used CSI-Dlv (Cronbach’s alpha = 0.91, test-retest reliability = 0.88 for patients with 112 chronic pain and 0.91 for healthy volunteers)[8] showed good internal consistency”

Please rephrase and include only useful information.

-It is not clear to me whether someone was double-checking the part B reported by the patients. I think patients would have some difficulties in correctly filling in that part, as actually reflected by the fact that 1036 out of 4739 screened patients were excluded due to the lack of part B (Flowchart figure 1).

To me this Is an important point and should be clarified in the text as well, since as the authors state, the reported CSSs is based on the information provided in the CSI part B.

RESULTS

- first paragraph lines 219-222 : please include also percentages for the women/men numbers.

- Figure 2: is too small and can’t be read. Maybe you can rotate it and make it a full-page image?

-The same for table 1: too much information and is difficult to follow.

DISCUSSION

-Rather than starting the discussion with a statement on the main aims, I’d suggest reporting the main results of the article.

Minor editing required

Reviewer 3 Report

The authors have thoroughly and impressively achieved their primary aim which was to establish the cut-off values for the CSI-Dlv in patients with chronic pain (total group and per sex) with one or more CSSs and healthy volunteers as comparisons. 

Their secondary aims were (1) to address differences between groups (patients with chronic pain and healthy volunteers, males and females and the presence or absence of CSS) and (2) to address whether patient descriptives, pain severity, pain catastrophising, pain disability, number of pain locations, number of reported CSSs and health-related quality of life are factors associated with the CSI score. It would be helpful to the reader to state in the Conclusions the extent to which the secondary aims were achieved. It requires intensive reading to be satisfied that the secondary aims were achieved.

Promotion of the term “human assumed central sensitization” (HACS) is a good idea.

I searched for meaning and perspective about the CSI (and I don`t mean a comprehensive review of measures which suggest CSI may be a valid measure of human assumed central sensitization or a similar construct, as that would be beyond the scope of this study. Rather perhaps a paragraph which may be helpful for a naïve reader.) To what extent does the CSI indicate anything more than the association with a group of clinical conditions (CSS) in which there has been evidence suggestive of central sensitization (HACS)? I note the heterogeneous factors which highly significantly influence the CSI score: sex, age, pain catastrophizing, pain disability, number of pain locations, number of reported CCSs, physical and mental components of the SF-12 (quality of life). Would the authors care to define exactly what the CSI measures or assesses?

Some studies such as reference [68] do give indications that the CSI is giving some insight into HACS. However, the systematic review and meta-analysis by Adams et al was cited [78] without comment but merits more consideration: “The CSI and PSQ (Pain Sensitivity Questionnaire) showed weak or no correlations with nociceptive sensitivity: pain thresholds, temporal summation, or conditioned pain modulation.” Rather, they found that the CSI was strongly correlated with psychological measures (anxiety, depression, pain catastrophising, stress, sleep, and kinesiophobia). Shraim et al (Methods to discriminate between mechanism-based categories of pain experienced in the musculoskeletal system: a systematic review. Pain. 2021 Apr 1;162(4):1007-1037) make the point that “Few methods have been validated for discrimination between mechanism-based categories of pain.” Rather, a combination of features and methods was recommended to discriminate between the categories of musculoskeletal pain, including nociplastic pain. It is acknowledged that the CSS group differ from musculoskeletal pain, except when nociplastic features are prominent, including so-called secondary fibromyalgia. Multisensory sensitivity (Wang D, Frey-Law LA. Multisensory sensitivity differentiates between multiple chronic pain conditions and pain-free individuals. Pain. 2023 Feb 1;164(2):e91-e102.), as assessed by the Multisensory Amplification Scale, is not a measure of pain central sensitization specifically, rather it represents a related construct of generalised central sensitivity and/or risk factor for altered CNS processing and may be advantageous in that it only correlates weakly with psychological measures. 
